# Bad Global Minima Exist and SGD Can Reach Them

**Shengchao Liu**
Quebec Artificial Intelligence Institute (Mila)
Université de Montréal
liusheng@mila.quebec

**Dimitris Papailiopoulos**
University of Wisconsin-Madison
dimitris@papail.io

**Dimitris Achlioptas**
University of Athens
optas@di.uoa.gr

## Abstract

Several works have aimed to explain why overparameterized neural networks generalize well when trained by Stochastic Gradient Descent (SGD). The consensus explanation that has emerged credits the randomized nature of SGD for the bias of the training process towards low-complexity models and, thus, for implicit regularization. We take a careful look at this explanation in the context of image classification with common deep neural network architectures. We find that if we do not regularize *explicitly*, then SGD can be easily made to converge to poorly-generalizing, high-complexity models: all it takes is to first train on a random labeling on the data, before switching to properly training with the correct labels. In contrast, we find that in the presence of explicit regularization, pretraining with random labels has no detrimental effect on SGD. We believe that our results give evidence that explicit regularization plays a far more important role in the success of overparameterized neural networks than what has been understood until now. Specifically, by penalizing complicated models independently of their fit to the data, regularization affects training dynamics also far away from optima, making simple models that fit the data well discoverable by local methods, such as SGD.

## 1 Introduction

In [1], Zhang et al. demonstrated that several popular deep neural network architectures for image classification have enough capacity to perfectly memorize the CIFAR10 training set. That is, they can achieve zero training error, even after the training examples are relabeled with uniformly random labels. Moreover, these memorizing models are not even hard to find; they are reached by standard training methods such as stochastic gradient descent (SGD) in about as much time as it takes to train with the correct labels. It would stand to reason that since these architectures have enough capacity to "fit anything," models derived by fitting the correctly labeled data, *i.e.*, "yet another anything," would fail to generalize. Yet, miraculously, they do not: models trained by SGD on even severely overparameterized architectures generalize well. Following recent work [2, 3, 4, 5, 6, 7, 8, 9, 10, 11, 12], our study is motivated by the desire to shed some light onto this miracle which stands at the center of the recent machine learning revolution.

When the training set is labeled randomly, all models that minimize the corresponding loss function are equivalent in terms of generalization, in the sense that, we expect none of them to generalize. The first question we ask is: when the true labels are used, are *all* minima of the loss function equivalent in terms of generalization, or are some better than others? As we see—perhaps unsurprisingly—not all global minima are created equal: there exist bad *global* minima, *i.e.*, global minima that generalize

poorly. This is compatible with the findings of the experiments in [13], that generate bad global minima in a similar way as we do, but with less of a dramatic drop in test accuracy.

The existence of bad global minima is rather unsurprising, but implies something important: the optimization method used for training, *i.e.*, for selecting among the different (near-)global minima, has *germane* effect on generalization. In practice, SGD appears to avoid bad global minima, as different models produced by SGD from independent random initializations tend to all generalize equally well, a phenomenon attributed to an inherent bias of the algorithm to converge to models of "low complexity" [14, 15, 16, 17, 18, 19]. This brings about our second question: does SGD deserve all the credit for avoiding bad global minima, or are there also other factors at play? More concretely, can we initialize SGD so that it ends up at a bad global minimum? Of course, since we can always start SGD *at* a bad global minimum, our question has a trivial positive answer as stated. We show that initializations that cause SGD to converge to bad global minima can be constructed given only *unlabeled* training data, *i.e.*, without knowledge of the true loss landscape.

The fact that we can construct bad initializations without knowledge of the loss landscape suggests strongly that these initializations correspond to models with *inherently* undesirable characteristics which persist, at least partially, in the trained models that fit the correct labels. Such a priori undesirability justifies a priori preference of some models over others, *i.e.*, regularization. In particular, if a regularization term makes such models appear far worse than before, this correspondingly incentivizes SGD to move away from them when initialized at the models. This is precisely what we find in our experiments: adding $l_2$ regularization and data augmentation, *i.e.*, regularization that favors models invariant to the transformations used to augment the data, allows SGD to overcome the effect of pretraining with random labels and end up at good global minima. In that sense, data augmentation and regularization appear to play a very significant, and largely unexplored, role beyond distinguishing between different models that fit the data equally well, *e.g.,* as studied in [20]. They in fact affect training dynamics *far away from optima,* making good models easier to find, perhaps by making bad models more evidently bad.

**A Sketch of the Phenomenon**  As an illustrative toy-example, we consider the task of training a two-layer, fully-connected neural network for binary classification, where the training data is sampled from two identical, well-separated 2-dimensional Gaussian distributions. In our example, each class comprises 50 samples, while the network has 100 hidden units in each layer and uses ReLU activations. In Figure 1, we show the decision boundary of the model reached by training with SGD until 100% accuracy is achieved, in four different settings:

1. Random initialization + Training with true labels.

2. Random initialization + Training with random labels.

3. Random initialization + Training with random labels + Training with true labels.

4. Random initialization + Training with random labels + Training with true labels using data augmentation[1] and $l_2$ regularization.

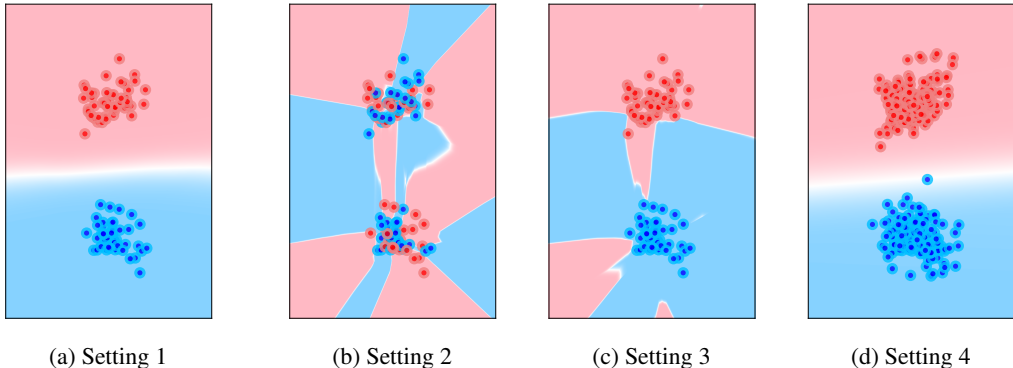

| (a) Setting 1 | (b) Setting 2 | (c) Setting 3 | (d) Setting 4 |

Figure 1: The decision boundary of the model reached by SGD in Settings 1–4, respectively.

Figure 1a shows that from a random initialization, SGD converges to a model with near max margin. Figure 1b shows that when fitting random labels, the decision boundary becomes extremely complex and has miniscule margin. Figure 1c shows that when SGD is initialized at such an extremely complex model, it converges to a "nearby" model whose decision boundary is unnaturally complex and has small margin. Finally, in Figure 1d, we see that when data augmentation and regularization are added to the training regime, SGD manages to escape the bad initialization corresponding to first training with random labels and, again, reaches a model with a max margin decision boundary.

In Section 3, we show that the phenomenon sketched above persists in state-of-the-art neural network architectures over real datasets. Specifically, we examine VGG16, ResNet18, ResNet50, and DenseNet40, trained on CIFAR, CINIC10, and a restricted version of ImageNet. In all cases, we find the following: 1) pretraining with random labels causes subsequent SGD training on true labels to fail, *i.e.*, when started from a model trained to fit random labels, SGD finds models that fit the true labels perfectly but have poor test performance, and 2) adding regularization to the training on true labels (either explicit or as data augmentation) allows SGD to overcome the bad initialization caused by pretraining with random labels and converge to models with good test performance.

## 2 Experimental Setup

**Datasets and Architectures**   We ran experiments on the CIFAR [21] dataset (including CIFAR10 and CIFAR100), CINIC10 [22] and a resized Restricted ImageNet [23].[2] The CIFAR training set consists of 50k data points and the test set consists of 10k data points. The CINIC10 training set consists of 90k data points and the test set consists of 90k data points. The Restricted ImageNet training set consists of approximately 123k data points and the test set consists of 4.8k data points. We train four models on them: VGG16 [24], ResNet18 and ResNet50 [25] and DenseNet40 [26].

**Implementation and Reproducibility**   We run our experiments on PyTorch 0.3. Our figures, models, and all results can be reproduced using the code available at an anonymous GitHub repository: `https://github.com/chao1224/BadGlobalMinima`.

**Training methods**   We consider the state-of-the-art (SOTA) SGD training and vanilla SGD training. The former corresponds to SGD with data augmentation (random crops and flips), $\ell_2$-regularization, and momentum. The latter corresponds to SGD without any of these features.

---

**Algorithm 1** Adversarial initialization

---

**Input:** Original training dataset $S$; Replication factor $R$; Noise factor $N$
$C = \emptyset$
**for** every image $x \in S$ **do**
    **for** $i$ from 1 to $R$ **do**
        $x_i \leftarrow$ zero-out a random subset comprising $N\%$ of the pixels in $x$
        $y_i \leftarrow$ Uniformly random label
        Add $(x_i, y_i)$ to $C$
    **end for**
**end for**
Train the architecture to $100\%$ accuracy on $C$ from a random initialization using vanilla SGD [3]
**Output:**   The weight vector of the architecture when training ends

---

**Initialization**   We consider two kinds of initialization: random and adversarial, *i.e.*, generated by training on a random labeling of the training data. For random initialization we use the PyTorch defaults. Specifically, both the convolutional layers and fully connected layers are initialized uniformly. To create an adversarial initialization, we start from a random initialization and train the model on an augmented version of the original training dataset where we have labeled every example uniformly at random. All details can be found in Algorithm 1.[4]

**Hyperparameters**    We apply well-tuned hyperparameters for each model and dataset. For CIFAR, CINIC10, and Restricted ImageNet, we use batch size 128, while the momentum term is set to 0.9 when it is used. When we use $\ell_2$ regularization, the regularization parameter is $5 \cdot 10^{-4}$ for CIFAR and Restricted ImageNet and $10^{-4}$ for CINIC10. We use the following learning rate schedule for CIFAR: 0.1 for epochs 1 to 150, 0.01 for epoch 151 to 250, and 0.001 for epochs 251 to 350. We use the following learning rate schedules for CINIC10 and Restricted ImageNet: 0.1 for epochs 1 to 150, 0.01 for epoch 151 to 225, and 0.001 for epochs 226 to 300.

Before we proceed, we would like to note that one can potentially force vanilla SGD to escape bad initializers (or any initializers for that matter) by employing non-standard, adaptive learning schedules such as those in [27, 28], or by using "unnatural", or enormously large learning rate for a few iterations and then switching back to a more standard-practice schedule. The goal of our experiments is to isolate the effects of explicit regularization and implicit bias—to the extent possible—from that of the choice of learning rates. Hence, across all experiments, we adopt standard-practice, decaying schedules, optimized for good convergence speeds.

## 3    Experimental Findings and Observed Phenomena

The motivation behind our initializations comes from the expectation that the need to memorize random labels will consume some of the learning capacity of the network, reducing the positive effects of overparameterization. Furthermore, as seen in the toy example in Section 1, training on random labels yields complex decision boundaries. As a result, the question becomes whether the bias of SGD towards simple models suffices to push it away from the initial (complex) model, or not. We find that it does not suffice. In contrast, we find that the bias towards simple models induced by regularization and data augmentation, does.

To offer insight on our above core finding, in the following we present several metrics related to the trained models. To generate the experimental results, we ran each setup 5 times with different random seeds and reported the min, max and average accuracy and complexity for each model and dataset.

We first report the train and test accuracy curves for our 16 setups (4 datasets and 4 models), followed by the impact of the replication parameter $R$ on the test accuracy. We then examine how different combinations of training heuristics (*e.g.*, data augmentation, regularization and momentum) impact the overall test accuracy with and without pretraining on random labels. After this, we report the distance that a model travels, first from a random initialization to the model that fits random labels and then from that model to the model trained on the correct labels. Finally, we report several norms that are well-known proxies for model complexity, and also the robustness of all models trained against adversarial perturbations, *i.e.*, yet another model complexity proxy.

Our main observations as taken from the figures below and our experimental data are as follows:

1. Vanilla SGD on the true labels from a random initialization reaches 100% training accuracy for all models and datasets tested, which is consistent with [1].
2. Vanilla SGD on the true labels after first training on random labels reaches 100% training accuracy, but suffers up to 40% test accuracy degradation compared to the model reached from a random initialization. That is, models which fit the true training labels perfectly can have a difference of up to 40% in test accuracy: not all global optima are equally good.
3. SOTA SGD converges to nearly the same test accuracy from random vs. adversarial initialization, *i.e.*, it escapes the detrimental effect of first training on random labels.
4. Data augmentation, $\ell_2$ regularization and momentum all help SGD to move far away (in euclidean distance) from adversarial initializations; in sharp contrast, vanilla SGD finds a minimizer close to the adversarial initialization.

### 3.1    Training Accuracy

In Figure 2 we report the training accuracy convergence for all models setups on all four datasets. We consistently observe two phenomena.

First, we see that after a sufficient number of epochs for most setups, we can reach 100% training accuracy irrespective of the initialization, or the use of data augmentation, regularization or momentum. The only outlier here would be DenseNet since it seems to only achieve a little less than 100%

training accuracy on some datasets. However, in the vast majority of settings the models can perfectly fit the training data.

The second phenomenon observed is that the speed of convergence to 100% training accuracy is irrespective of whether we start with adversarial or random initialization. This may hint at the possibility that models that fit the true labels perfectly (which exist for most cases) are close to most initializations, including the adversarial one.

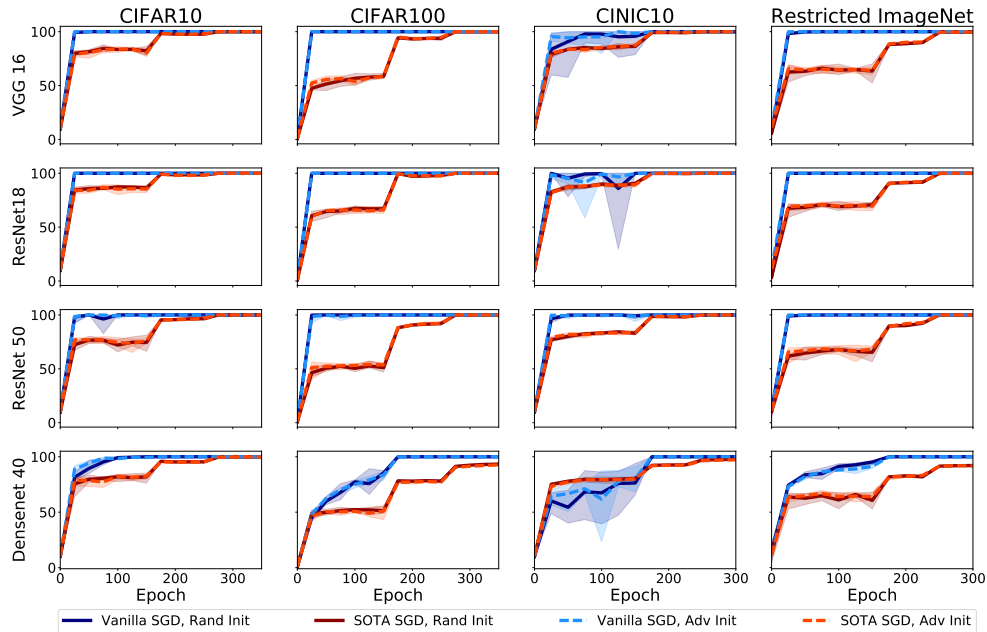

Figure 2: Training accuracy (%) vs number of epochs on CIFAR, CINIC10 and Restricted ImageNet on all four neural network models.

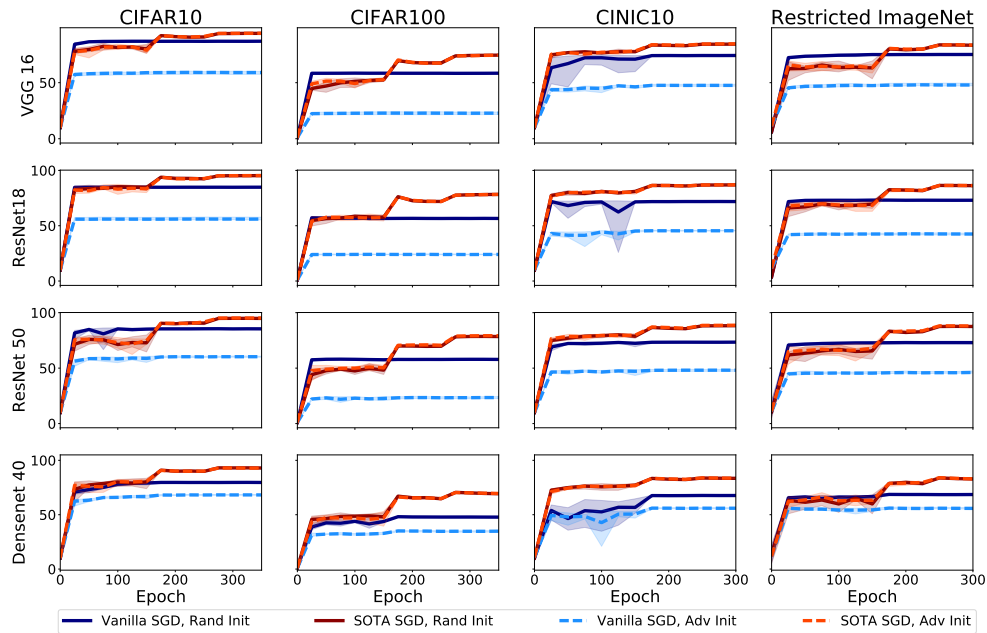

Figure 3: Test accuracy (%) vs number of epochs on CIFAR, CINIC10 and Restricted ImageNet on all four neural network models.

## 3.2 Test accuracy

Our most important findings are shown in Figure 3, which illustrates the test accuracy convergence during training. We see that the test accuracy of adversarially initialized vanilla SGD flattens out significantly below the corresponding accuracy under a random initialization, even though both methods achieve 100% training accuracy. The test accuracy degradation can be up to 40% on CIFAR100, while for DenseNet the test degradation is comparatively smaller.

At the same time, we see that the detrimental effect of adversarial initialization vanishes once we use data augmentation, momentum, and $\ell_2$ regularization. This demonstrates that by also changing the optimization landscape *far* from good models, the above heuristics play a role that has not received much attention, namely effecting the *dynamics* of the search for good models.

It is natural to ask if the bad global models to which SGD converges when adversarially initialized have some shared properties. In the following, we argue that one property that stands out is that such bad global minima are in a small neighborhood within the adversarial initializers; and, it appears, that as long as it can find a perfect fit, SGD prefers these models, as it take less "effort" to converge to. In contrast, as we see later on, SOTA SGD forces travel far from the bad initialization.

## 3.3 Distance Travelled during Training

Here we report on the distance travelled from an initializer till the end of training in our different setups. First we define the distance between the parameters of two models $W_1$ and $W_2$ as $d(W_1, W_2) = \frac{\|W_1 - W_2\|_F}{\|W_2\|_F}$, where $\|\cdot\|_F$ denotes the Frobenius norm.

In Table 1, we report the distance travelled from a random vs. an adversarial initialization to the final model (achieving 100% training accuracy). A subscript 0 denotes an initializer; an $S$ or $V$ subscript indicates training with SOTA or vanilla SGD. The $R$ or $A$ superscripts indicate whether the model has been initialized randomly or adversarially.

We observe that from a random initialization, the distance that vanilla SGD travels to a model with 100% training accuracy is independent of whether we use the correct labels or random labels. Specifically, whether we want to train a genuinely good model, or to find an adversarial initialization the distance traveled is about 0.9. Intriguingly, when vanilla SGD is initialized adversarially, the distance to a model with 100% training accuracy on the correct labels is far less than 0.9, being approximately 0.2. This can be interpreted as SGD only spending a modicum of effort to "fix up" a bad model, just enough to fit the training labels.

In contrast, when training with the correct labels, the distance travelled by SOTA SGD to a model with 100% training accuracy is significantly larger if we initialize adversarially vs. if we initialize randomly, in most cases by nearly an order of magnitude. This hints at the possibility that data augmentation, regularization and momentum enable SGD to escape the bad initialization by dramatically altering the landscape in its vicinity (and beyond).

Finally, we observe that bad models seem to always be in close proximity to random initializers, *i.e.*, bad models are easy to find from almost every point of the parameter space.

Table 1: The model distance (mean of 5 random runs) for the different datasets and models.

| Dataset | Model | $d(W_0^R, W_V^R)$ | $d(W_0^R, W_S^R)$ | $d(W_0^R, W_0^A)$ | $d(W_0^R, W_V^A)$ | $d(W_0^R, W_S^A)$ | $d(W_0^A, W_V^A)$ | $d(W_0^A, W_S^A)$ |
|---|---|---|---|---|---|---|---|---|
| CIFAR10 | DenseNet40 | 0.810 | 3.715 | 0.946 | 0.950 | 3.715 | 0.347 | 28.669 |
| | ResNet18 | 0.953 | 3.917 | 0.873 | 0.879 | 3.902 | 0.207 | 12.421 |
| | ResNet50 | 0.894 | 8.552 | 0.919 | 0.923 | 8.608 | 0.194 | 36.311 |
| | VGG16 | 0.907 | 3.464 | 0.950 | 0.953 | 3.433 | 0.264 | 26.838 |
| CIFAR100 | DenseNet40 | 0.917 | 2.008 | 0.946 | 0.968 | 1.980 | 0.538 | 13.290 |
| | ResNet18 | 0.915 | 2.609 | 0.852 | 0.862 | 2.597 | 0.210 | 7.243 |
| | ResNet50 | 0.917 | 4.941 | 0.925 | 0.927 | 4.882 | 0.172 | 24.598 |
| | VGG16 | 0.919 | 2.029 | 0.960 | 0.962 | 2.035 | 0.253 | 19.388 |
| CINIC10 | DenseNet40 | 0.917 | 1.415 | 0.955 | 0.976 | 1.440 | 0.491 | 12.102 |
| | ResNet18 | 0.836 | 1.565 | 0.942 | 0.951 | 1.560 | 0.212 | 10.961 |
| | ResNet50 | 0.837 | 3.260 | 0.952 | 0.955 | 3.274 | 0.157 | 24.940 |
| | VGG16 | 0.844 | 1.406 | 0.974 | 0.977 | 1.397 | 0.241 | 17.916 |
| Restricted ImageNet | DenseNet 40 | 0.940 | 3.378 | 0.966 | 0.982 | 3.377 | 0.520 | 39.314 |
| | ResNet 18 | 0.849 | 3.154 | 0.969 | 0.973 | 3.164 | 0.163 | 47.184 |
| | ResNet 50 | 0.883 | 7.123 | 0.913 | 0.916 | 7.139 | 0.164 | 30.841 |
| | VGG 16 | 0.836 | 2.885 | 0.985 | 0.986 | 2.871 | 0.172 | 83.035 |

### 3.4 The Effect of the Replication Factor $R$ on Test Accuracy

Here, we report the test accuracy effect of the replication factor $R$, *i.e.*, the number of randomly labeled augmentations that are applied to each point during adversarial initialization. In Figure 4, we plot the test accuracy performance for all networks we tested as a function of the number of the randomly labeled augmentations $R$. When we vary $R$, we observe that SOTA SGD essentially achieves the same test accuracy, while the test performance of vanilla SGD degrades, initially fast, and then slower for larger $R$.

We would like to note that although it would be interesting to make $R$ even bigger, the time needed to generate the adversarial initializer grows proportional to $R$, as it requires training a dataset (of size proportional to $R$) to full accuracy.

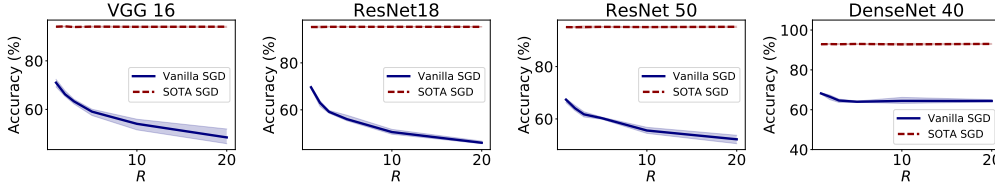

Figure 4: The effect of $R$ on CIFAR10, the zero-out ratio is fixed to $10\%$. Clearly, increasing $R$ causes vanilla SGD to suffer more. In contrast, SOTA SGD always stays unaffected.

### 3.5 The Effect of Different Training Heuristics

In our implementation, SOTA SGD involves the simultaneous use of data augmentation, $\ell_2$ regularization and momentum. Here we tease out the different effects, by exploring all 8 combinations of the 3 heuristics as shown in Table 2, in order to examine if a specific heuristic is particularly effective at repairing the test accuracy damage done by an adversarial initialization. What we find is that each heuristic by itself is enough to allow SGD to largely escape a bad initialization, but not to reach the same level of test accuracy as from a random initialization. When a second heuristic is added, though, the test accuracy becomes independent of the initialization, in all three combinations.

Table 2: Multiple SGD results with random init and adversarial init respectively using model ResNet18 on CIFAR10.

| | Random Init | | Adversarial Init | |
|---|---|---|---|---|
| Mode | Train Acc | Test Acc | Train Acc | Test Acc |
| Vanilla SGD | $100.000 \pm 0.000$ | $84.838 \pm 0.193$ | $100.000 \pm 0.000$ | $56.024 \pm 0.883$ |
| DA | $100.000 \pm 0.000$ | $93.370 \pm 0.115$ | $99.995 \pm 0.003$ | $89.402 \pm 0.175$ |
| $\ell_2$ | $100.000 \pm 0.000$ | $87.352 \pm 0.055$ | $100.000 \pm 0.000$ | $83.172 \pm 3.732$ |
| Momentum | $100.000 \pm 0.000$ | $89.200 \pm 0.176$ | $100.000 \pm 0.000$ | $89.016 \pm 0.142$ |
| DA+$\ell_2$ | $100.000 \pm 0.000$ | $94.680 \pm 0.071$ | $100.000 \pm 0.000$ | $94.192 \pm 0.173$ |
| DA+Momentum | $100.000 \pm 0.000$ | $93.448 \pm 0.242$ | $100.000 \pm 0.001$ | $92.756 \pm 0.347$ |
| $\ell_2$+Momentum | $100.000 \pm 0.000$ | $89.050 \pm 0.110$ | $100.000 \pm 0.000$ | $88.932 \pm 0.373$ |
| DA+$\ell_2$+Momentum (SOTA) | $100.000 \pm 0.000$ | $95.324 \pm 0.086$ | $100.000 \pm 0.001$ | $95.346 \pm 0.098$ |

### 3.6 Proxies for Model Complexity

Here we report on some popular proxies for model complexity, *e.g.*, the Frobenius norm of the weights of the network, and recently studied path norms [2]. For brevity, we only report the results using ResNet50, while all remaining figures can be found in the supplemental material.

We observe that across all three norms, the network tends to have smaller norm for SOTA SGD irrespective of the initialization. At the same time, we observe that the adversarially initialized model trained with vanilla SGD has larger norms compared to random initialization.

Taking these norms as proxies for generalization, then they indeed do track our observations that adversarial initialization leads vanilla SGD to worse generalization, and SOTA SGD explicitly biases towards good global models, while being unaffected by adversarial initialization.

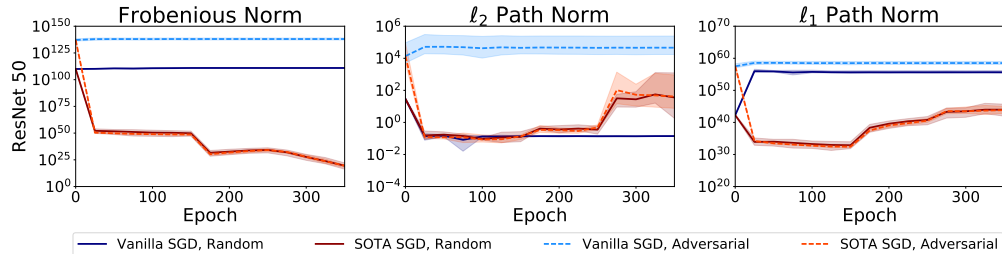

Figure 5: Norm measurement on CIFAR10 and ResNet 50.

## 3.7 Robustness to Adversarial Examples

Robustness to adversarial examples is another metric we examine. In this case, model robustness is a direct proxy for margin, *i.e.*, the proximity of the decision boundary to the train, or test data. In Figure 6, we report the test accuracy degradation once adversarial perturbations are crafted for Resnet50 on all four datasets. To compute the adversarial perturbation we use the Fast Gradient Sign Attack (FGSM) [11] on the final model.

Consistent with the norm measures in the previous subsection, we observe that when adversarially initialized, vanilla SGD finds models that are more prone to small perturbations that lead to misclassification. This is to be expected, since (as also observed in the toy example), the decision boundaries for adversarially initialized vanilla SGD tend to be complex, and also are very close to several training points (*i.e.*, their margin is small).

As observed in the previous figures, the decision boundaries of models derived by SOTA SGD are less prone to adversarial attacks, potentially due to the fact that they achieve better margin.

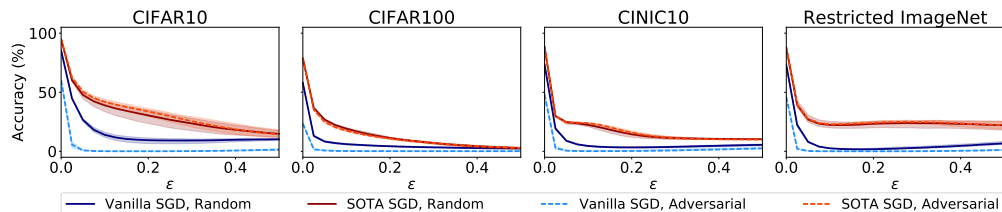

Figure 6: FGSM on CIFAR, CINIC10 and Restricted ImageNet using ResNet50.

## 4 Conclusion

Understanding empirical generalization in the face of severe overparameterization has emerged as a central and fascinating challenge in machine learning, primarily due to the dramatic success of deep neural networks. Several studies aim to explain this phenomenon. Besides "no bad local minima", the emergent consensus explanation is that SGD is biased towards minima of low complexity.

In this work, we first show that models that fit the training set *perfectly* yet have poor generalization not only exist, but are what SGD converges to if training on the true labels is preceded by training on random labels. In other words, the bias of SGD towards simple models is not enough to overcome the effect of starting at a complicated model (that fits random labels). Then we show that, in contrast to the above picture, regularization (either explicit or in the form of data augmentation) and momentum, do suffice for SGD to escape the effect of the initialization and reach models that generalize well.

## Broader Impacts

We believe that the main value of our work is in pointing out the crucial, yet largely unexplored, role played by regularization in *search dynamics*. This is a departure from the usual way of thinking about generalization, wherein its role is to ensure stability of the chosen model with respect to fluctuations

in the training sample. In other words, we make the point that regularization is important not only in its role of demoting (penalizing) complex models that fit the data well, but even in penalizing complex models that fit the data poorly, altering the entirety of the optimization landscape and making the space of simple models better searchable by local methods such as SGD. We understand that our work leaves open the exact mechanism through which regularization achieves this effect, but the experimental evidence we give for this effect is undeniable.

Given the enormous intellectual importance of regularization in machine learning, the possibility that its role is actually far larger in scope than previously realized, is rather remarkable. In that sense, the value of our work is in opening up a potentially large domain of further research, namely understanding the role of regularization in search dynamics, including the possibility of a future direction wherein regularization is aimed not only at promoting model stability but also model discoverability.

## Acknowledgements

Dimitris Papailiopoulos is supported by an NSF CAREER Award #1844951, two Sony Faculty Innovation Awards, an AFOSR & AFRL Center of Excellence Award FA9550-18-1-0166, and an NSF TRIPODS Award #1740707.

## Footnotes

[1]Data augmentation was performed by replicating each training point twice and adding Gaussian noise.

[2]We resize the images from Restricted ImageNet to $32 \times 32$ for faster computation, which leads to a test accuracy drop from 96% to 87%.

[4]For architectures that cannot reach 100% accuracy on random labels we train until convergence occurs.

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
