[Reviews · NeurIPS 2020]

Review 1

Summary and Contributions: The paper shows that networks trained first on random labels, then on correct labels, do not generalize well when trained using vanilla SGD: they converge to bad global minima. However, when the same networks are trained using any combination of momentum, data augmentation, and L2 regularization, do learn a good solution (a good global minimum) despite the initial, adversarial training.

Strengths: The paper presents a solid empirical case for the claims made above. The results are established on many combinations of network architectures and datasets. The paper supports the main conclusion with an ablation study showing separately the effect of each regularization method on the training performance. The paper also illustrates the effect on a simple 2D dataset.

Weaknesses: While the paper shows an intuitive example based on 2D samples where the phenomenon occurs, it does not offer an explanation as to how different regularization methods help SGD avoid bad global minima. Since this phenomenon does occur in the simple 2D setting, this seems like a good starting point for attempting to answer this question.

Correctness: I believe the claims made in the paper. The effect is demonstrated across many combinations of architectures and datasets, and the experimental setup is clearly explained and well thought-out.

Clarity: The paper is extremely well written.

Relation to Prior Work: The paper attempts to explore the conditions under which SGD is able to find good global minima. As such, it refers to previous works which attempted to explain how SGD achieves this feat. In contrast, the present work shows the limits of SGD, and that its performance is greatly aided by good initialization and/or by regularization.

Reproducibility: Yes

Additional Feedback:


Review 2

Summary and Contributions: This paper presents and amplifies an important idea: that not all global minima are equal from a generalization perspective. They provide an empirical construction to create initializations (adversarial initializations) from which SGD training trajectories end up at minima that generalize poorly compared to trajectories initialized from random initializations. A second contribution of the paper is to show that in the presence of explicit regularization SGD can overcome these adversarial initializations, and thus explicit regularization in a sense easily overcomes the implicit bias of SGD. Although the paper is positioned more along the lines of the second contribution, I find the first one to be the more interesting one -- I believe it really helps shed light on why SGD fundamentally (i.e., when there is no explicit regularization) generalizes.

Strengths: As mentioned above, I believe the novel empirical construction presented in this paper for adversarial initialization provides a very important insight into why (vanilla) SGD generalizes. I also believe this work has been quite influential in shaping the research in this area since the arXiv version first came out last year. For example, the theory of Coherent Gradients [a, b] which presents an intuitive explanation for why vanilla SGD generalizes (i.e., the question that motivated this work) was significantly influenced by the arXiv version of this work. [a] https://arxiv.org/abs/2002.10657 (ICLR 2020) [b] https://arxiv.org/abs/2003.07422

Weaknesses: There are a few aspects of the empirical methodology that I would like to see explored more thoroughly. These basically all center around how fragile is the construction of adversarial initializations. (Of course, it is interesting that a construction exists at all, so these are secondary questions in a certain sense.) (1) In Algorithm 1, how crucial is the zeroing out of the random subset of pixels? It seems a little unnatural (for example in other applications there may not be a natural analog of zeroing out), so it would be good to see what happens if that is removed. And if it turns out necessary, would I would like to understand why. (2) Experiments to more carefully control the learning rate schedule would be very useful. I worry that a minimum found with a low learning rate (towards the end of training with random labels) may be easily lost when training on real labels (which starts with a higher learning rate). In the write up, please explain what happens to the learning rate schedule in the course over the full picture i.e. random init --- adversarial training ---> adversarial init ---- real training ---> end and if possible would like to see results of what happens if we keep the learning rate constant.

Correctness: I believe so, modulo the issues/clarification pointed out above in the limitations.

Clarity: Yes, the paper is well written (but would like more discussion around the points noted in the limitations).

Relation to Prior Work: There is an important line of related work in the theoretical community which shows that different optimizers (e.g., SGD and ADAM) may reach minima with different generalization characteristics [c, d] through the means of simple analytical examples. I believe these should be discussed. And although it is not prior work, it would be interesting to get the authors take on Coherent Gradients [a, b] as a possible explanation for their observations. [c] https://arxiv.org/abs/1705.08292 [d] https://arxiv.org/abs/1811.07055

Reproducibility: Yes

Additional Feedback: --- Update after Rebuttal --- Thank you for the response. I look forward to the paper updated with the related work pointed out by the reviewers and the new experiments.


Review 3

Summary and Contributions: This paper empirically studies the optimization landscape of deep neural networks by investigating the performance of SGD around artificially constructed bad initializations. The main contributions of this paper are: -- Proposed an adversarial initialization technique, consisting of training on corrupted labels till 100% training accuracy. -- Empirically showed that when initialized adversarially, unregularized vanilla SGD generalizes much poorer than from random initialization, whereas the SOTA SGD (with regularization, data augmentation, momentum) can recover from adversarial initialization and reach the same performance as from random init. -- Provided experimental understandings on why and how SOTA SGD is better than vanilla SGD through distance to initialization, norm-based capacity measure, and adversarial robustness.

Strengths: -- The core observation of SOTA SGD can recover from bad initialization, whereas vanilla SGD cannot, is very interesting and could be of broad interest to both practitioners and theoreticians. On the practical side, this observation replies to the “Understanding Generalization” paper by Zhang et al. and resurrects the importance of proper regularization. As an example of the implications, for newly designed deep architectures with not necessarily good optimization landscapes, perhaps explicit regularization is still necessary. -- The technique for getting the bad initialization (through training on corrupted labels) is natural (e.g. given Zhang et al.) but quite neat and has the potential to be used a lot in future work. -- I find the diagnostic experiments (in Section 3.3-3.7) quite informative. A lot of these experiments confirm or agree with recent insights in deep learning theory, and these sections report concrete numbers (e.g. distance travelled; ablation on various components in SOTA SGD; norm-based capacities) that I imagine a good portion of future research could get insights from.

Weaknesses: -- My main concern about this work is the relative lack of discussion and situation in the literature. After reading the paper, I didn’t get a good understanding of whether similar things are tried in the literature on studying the optimization landscape / trajectory empirically. I noticed there are citations on Line 26 about prior work along these lines, but it’s in batch. I encourage the authors to discuss the difference between them and the present paper or maybe point out other similar work in the rebuttal. -- The paper does not have theoretical results. Given that the empirical results are quite interesting I don’t think this concern is very major, but I am curious have the authors thought about theoretical directions in explaining the phenomena found here, especially given the recent advances in the understanding of the optimization (e.g. through the neural tangent kernels) and generalization of neural networks.

Correctness: The empirical methodologies in this paper are correct upon my inspection. The main experiments are comprehensive enough (4 architectures on 4 relatively large image tasks.) The diagnostic and ablation studies are well designed and have provided a more complete picture about the phenomena.

Clarity: The presentation of this paper is quite clear in general and I didn’t have a hard time understanding this paper.

Relation to Prior Work: As discussed above, I think the paper is a bit lacking in discussing the relation to prior work. -- There is no related work section. The citations in the introduction are mostly in batch without in-depth discussions on individual papers (except for [20]). -- There are no discussions about the relationship to recent advances in deep learning theory and understandings, especially the ones in the past one or two years.

Reproducibility: Yes

Additional Feedback:


Review 4

Summary and Contributions: This paper suggests that regularization helps avoiding SGD converging into bad global minima (ie. global minima with bad test accuracy).

Strengths: The paper introduces an interesting observation of (implicit) regularization. Previous works have shown that regularization does not help much to improve generalization in neural networks, and this paper shows that depending on the initialization regularization helps.

Weaknesses: - The paper claims to have shown for the first time that models that perfectly fit the training set can have different degrees of generalization depending on the initialization, ie. "global minima" with different test errors. This has been previously shown also using a similar technique. See for example "Theoretical issues in deep networks" by Poggio et al. (in PNAS), which shows (among other things) that depending on the standard deviation of the distribution to initialize the weights the network converges to global minima with different test accuracy (see Fig.2). Also, "Classical Generalization Bounds Are Surprisingly Tight For Deep Networks" by Liao et al. (CBMM Memo) introduces the training "Random initialization + Training with random labels + Training with true labels" and even more: they show that depending on the amount of images with randomized labels the test accuracy after training with the true labels varies accordingly (see Section 2). - For these other "bad global minima" found in previous works, would regularization help? - The paper does a poor job presenting the results: tons of redundant plots with irrelevant information (eg. Fig.2 and 3) and tables are hard to quickly extract conclusions (Table 1). - I would like to know more about the role of hyperparameters not related to regularization. The paper uses a SOTA SGD, but it is unclear if there is something in this SOTA SGD besides the regularization that could also potentially help to avoid "bad global minima". How are the hyperparameters (learning rates, batch size, momentums, etc.) adjusted? Maybe other configurations of these hyperparameters does not avoid "bad global minima"? -------POST REBUTTAL COMMENTS-------- Thank you for the clarification. I appreciate that the authors will cite previous work that already shows that bad global minima exist. However, the paper makes an immense claim (in the title "Explicit Regularization is Stronger than Implicit Bias") by providing evidence for 1 case of hyperparameters. While it is not possible to show empirically such claim universally, I would have expected at least to get sufficient information to know how to get to those parameters and be able to reproduce the experiments. I do not know how to reconcile the fact that hyperparameters were "tune[ed] for fast convergence" (in the rebuttal) and the weight decay and momentum hyperparameters are not 0 in the SOTA SGD given that vanilla SGD converges faster than with weight decay and momentum (blue lines vs red lines in the plots). So, I guess that the l2 regularizer and momentum are tuned with a different criterion, which one? (If the weight decay is too large the network won't fit the data and if it is too small it won't be able to forget the bad global minimum). I have dug into the code to see how the choice was made but that part of the code is not there. Then, once it is known how to reproduce the algorithm to choose the hyperparameters, it would be desirable to see at least a set of experiments with different choices of hyperparameters to support the main claim, given that there are other obvious hyperparameter choices that can challenge the claim, eg. - The learning rate alone could get the network out of the bad global minima: using a large learning rate after the adversarial training could help distort the parameters of the network and "forget" the bad global minima. After the network has escaped from the bad global minima, the learning rate could be reduced for convergence. Since the paper does not report results on different hyperparameters, it is not clear to me if it is the regularizer or the choice of the learning rate that is causing the avoidance of bad global minima. - The impact of having so many iterations in which the network is stuck in a plateau before decreasing the learning rate (see Fig. 2: for SOTA SGD there are plateaus due to too large learning rate while the vanilla SGD converges to global maximum): It could be that staying long in a sub-optimal plateau could help to avoid the bad global minima. In summary, this paper has serious problems of reproducibility and is missing important experimental controls to support the main claim. Also, I would suggest to re-design the figures to make the paper more readable.

Correctness: I am not sure as maybe other configurations of the hyperparameters (not related to regularization) are also responsible to avoid the "bad global minima".

Clarity: Yes, except the figures.

Relation to Prior Work: No. Previous works have shown that "bad global minima" exist.

Reproducibility: Yes

Additional Feedback:

[Author Response · NeurIPS 2020]

We thank all reviewers for their thoughtful feedback. We are encouraged by the overall positive assessment of our work
by Reviewers 1–3 (scores 7 7 6), who identified the novelty of our contribution, i.e, bringing to light the important
effect of regularization on SGD dynamics, and found that our claims are backed by informative experiments.

The biggest point of criticism, raised by **Reviewer 4**, concerns the novelty of our work. **R4** points to two papers, by
two closely related groups of authors:

[1] Poggio, Banburskia, Liao,　　　　　PNAS, June 2020.
[2] Liao, Miranda, Hidary, Poggio,　　CBMM Memo, July 2018.

The key message of our work is that regularization affects generalization not only by controlling capacity and, thus,
conferring stability, but also by its impact on **optimization dynamics:** without it SGD can converge to very poorly
generalizing models. **In contrast,** a main claim of [1,2] is that, in the *infinite* time limit and under smoothness
assumptions *not* satisfied by ReLU, regularization is unnecessary to SGD, as it converges to good, small norm solutions.
Besides the obvious clash, the only connection between [1,2] and our work is that their authors show that one can
achieve models with zero classification training loss but increasing amounts of test cross-entropy loss by either using
initial weights of increasing variance [1], or by pretraining with an increasing fraction of corrupted labels [2].

In our work we do indeed pretrain with corrupted labels, as in [2], and we will be happy to cite the, unknown to us,
CBMM memo in our work. That said, we need to state the following caveats:

1) Reference [2] pretrains with a *mixture* of corrupted and clean labels and shows this leads SGD to minima with worse
test **cross-entropy** (not classification) loss. The effect of the drop in test cross-entropy is moderate, i.e., $< 0.1$ drop; it is
difficult to interpret it with regards to classification loss. In contrast, our reported drop is with regards to actual test
**classification** loss and is dramatic, i.e., up to 40% degradation. In order to "bury" SGD so deeply:

21 　　• We use **completely**, not partially, corrupted labels.

22 　　• We employ **data augmentation** on the corrupted examples, i.e., we make several slightly different copies of
23 　　　each example and give it different random labels.

24 　　• We train to **full accuracy** on the corrupt data, not just for a fixed number of epochs as [1,2].

25 Each one of these differences is important in achieving the dramatic drop in generalization that we demonstrate.

26 2) **More importantly,** the majority of our paper is devoted into analyzing when/how/why SGD can "dig itself out" from
27 bad global minima. [1,2] **say nothing** on the main point of our paper, i.e., the role of regularization in SGD dynamics.

28 **R4** finally alludes that there may be other explanations for our results, e.g., hyperparameter tuning: *"it is unclear if*
29 *there is something [...] besides the regularization [...] to avoid "bad global minima". How are the hyperparameters [...]*
30 *adjusted?"* We challenge this: as reported, we do not perform any hyperparameter optimization to help SGD avoid, or
31 get stuck at bad initializers; we solely tune for fast convergence, precisely what is done in general. We find it peculiar
32 that **R4** both missed this and surmises "alternative explanations" in the face of our experimental data.

33 **R1**: *"[...] does not offer an explanation as to how different regularization methods help SGD avoid bad global minima."*
34 Although we don't include this in the paper (but are happy to do so), in the 2D case, all methods seem to equally help
35 SGD escape bad initializers. In the real-data case, none individually leads to SOTA accuracy, but combinations do. We
36 are very interested in understanding the individual effect of these methods, and are conducting relevant experiments.

37 **R2**: *"There are a few aspects [...] I would like to see explored more thoroughly. (1) [...] how crucial is the zeroing out*
38 *of the random subset of pixels? (2) Experiments to more carefully control the learning rate [...] would be very useful.*
39 (1) Zeroing-out is not crucial and a different technique could be used, e.g., additive noise on pixels, for which similar
40 effects can be observed. (2) Decreasing the LR won't affect the observations, but will lead to slower convergence, while
41 a larger LR won't give SOTA results. We are happy to include these additional experiments for different ranges of LR.
42 As suggested, we will discuss connections with coherent gradients. Thank you!

43 **R3**: *"My main concern about this work is the relative lack of discussion and situation in the literature."*
44 We will significantly expand the discussion on related work along the three most relevant threads: 1) "bad minima exist"
45 and ways to obtain them; 2) the effects of implicit bias; 3) the role of regularization. Due to lack of space here we can't
46 expand on all three above and related literature, but we will do so in the final version of this paper.

47 **R3**: *"The paper does not have theoretical results."*
48 Establishing theory to substantiate our findings would be thrilling but, frankly, these appear far beyond reach, for
49 any meaningful setting. Indeed, per [1], under sufficiently strong assumptions (infinite time, smooth losses), SGD
50 "converges" (in the sense of stationarity) to "wide" minima.

51 **R4**: *" For [...] "bad global minima" found in previous works, would regularization help? "*
52 We ran the experiments and indeed regularization helps massively. The adversarial initialization effect is completely
53 undone.

[Meta-Review · NeurIPS 2020]

3 of 4 reviewers reached a consensus that this paper deserves acceptance to neurips. Two additional feedbacks: (1) one of the reviewer thinks "Explicit Regularization is Stronger than Implicit Bias" is too strong of a claim, and the meta-reviewers agree that it might be better to make it "Explicit Regularization can be Stronger than Implicit Bias" (2) Explicit regualrization may trade with the learning rate when there is a batchnormlization involved. https://arxiv.org/abs/1910.07454. The paper should clarify or experiment with it.